# Generalizable One-shot 3D Neural Head Avatar

**Xueting Li, Shalini De Mello, Sifei Liu, Koki Nagano, Umar Iqbal, Jan Kautz**

NVIDIA

https://research.nvidia.com/labs/lpr/one-shot-avatar

## Abstract

We present a method that reconstructs and animates a 3D head avatar from a single-view portrait image. Existing methods either involve time-consuming optimization for a specific person with multiple images, or they struggle to synthesize intricate appearance details beyond the facial region. To address these limitations, we propose a framework that not only generalizes to unseen identities based on a single-view image without requiring person-specific optimization, but also captures characteristic details within and beyond the face area (*e.g.* hairstyle, accessories, *etc.*). At the core of our method are three branches that produce three tri-planes representing the coarse 3D geometry, detailed appearance of a source image, as well as the expression of a target image. By applying volumetric rendering to the combination of the three tri-planes followed by a super-resolution module, our method yields a high fidelity image of the desired identity, expression and pose. Once trained, our model enables efficient 3D head avatar reconstruction and animation via a single forward pass through a network. Experiments show that the proposed approach generalizes well to unseen validation datasets, surpassing SOTA baseline methods by a large margin on head avatar reconstruction and animation.

## 1 Introduction

Head avatar animation [57; 29; 38] aims to animate a source portrait image with the motion (i.e., pose and expression) from a target image. It is a long-standing task in computer vision that has been widely applied to video conferencing, computer games, Virtual Reality (VR) and Augmented Reality (AR). In real-world applications, synthesizing a realistic portrait image that matches the given identity and motion raises two major challenges – efficiency and high fidelity. Efficiency requires the model to generalize to arbitrary unseen identities and motion without any further optimization during inference. High fidelity demands the model to not only faithfully preserve intricate details in the input image (*e.g.* hairstyle, glasses, earrings), but also hallucinate plausibly whenever necessary (*e.g.* synthesize the occluded facial region when the input is in profile view or generate teeth when the mouth transitions from closed to open).

Traditional methods [14; 17; 11] based on 3D Morphable Models (3DMMs) learn networks that predict shape, expression, pose and texture of an arbitrary source portrait image efficiently. However, these approaches often fall short in synthesizing realistic details due to limited mesh resolution and a coarse texture model. Additionally, they exclusively focus on the facial region while neglecting other personal characteristics such as hairstyle or glasses. Inspired by the remarkable progress made in Generative Adversarial Networks (GANs) [22; 27; 58], another line of methods [57; 65; 67; 48; 46; 70; 15] represent motion as a warping field that transforms the given source image to match the desired pose and expression. Yet, without explicit 3D understanding of the given portrait image, these methods can only rotate the head within limited angles, before exhibiting warping artifacts, unrealistic distortions and undesired identity changes across different target views. Recently, neural rendering [39] has demonstrated impressive results in facial avatar reconstruction and

37th Conference on Neural Information Processing Systems (NeurIPS 2023).

animation [19; 42; 54; 55; 72; 2; 43; 20; 21; 23; 4]. Compared to meshes with fixed and pre-defined topology, an implicit volumetric representation is capable of learning photo-realistic details including areas beyond the facial region. However, these models have limited capacity and cannot generalize trivially to unseen identities during inference. As a result, they require time-consuming optimization and extensive training data of a specific person to faithfully reconstruct their 3D neural avatars.

In this paper, we present a framework aiming at a more practical but challenging scenario – given an unseen single-view portrait image, we reconstruct an implicit 3D head avatar that not only captures photo-realistic details within and beyond the face region, but also is readily available for animation without requiring further optimization during inference. To this end, we propose a framework with three branches that disentangle and reconstruct the coarse geometry, detailed appearance and expression of a portrait image, respectively. Specifically, given a source portrait image, our *canonical branch* reconstructs its coarse 3D geometry by producing a canonicalized tri-plane [9; 10] with a neutral expression and frontal pose. To capture the fine texture and characteristic details of the input image, we introduce an *appearance branch* that utilizes the depth rendered from the canonical branch to create a second tri-plane by mapping pixel values from the input image onto corresponding positions in the canonicalized 3D space. Finally, we develop an *expression branch* that takes the frontal rendering of a 3DMM with a target expression and a source identity as input. It then produces a third tri-plane that modifies the expression of the reconstruction as desired. After combining all three tri-planes by summation, we carry out volumetric rendering followed by a super-resolution block and produce a high-fidelity facial image with source identity as well as target pose and expression. Our model is learned with large numbers of portrait images of various identity and motion during training. At inference time, it can be readily applied to an unseen single-view image for 3D reconstruction and animation, eliminating the need for additional test-time optimization.

To summarize, our contributions are:

- We propose a framework for 3D head avatar reconstruction and animation that simultaneously captures intricate details in a portrait image while generalizing to unseen identities without test-time optimization.

- To achieve this, we introduce three novel modules for coarse geometry, detailed appearance as well as expression disentanglement and modeling, respectively.

- Our model can be directly applied to animate an unseen image during inference efficiently, achieving favourable performance against state-of-the-art head avatar animation methods.

## 2 Related Works

### 2.1 3D Morphable Models

Reconstructing and animating 3D faces from images has been a fundamental task in computer vision. Following the seminal work by Parke *et al*. [44], numerous methods have been proposed to represent the shape and motion of human faces by 3D Morphable Models (3DMMs) [1; 36; 16; 7; 35]. These methods represent the shape, expression and texture of a given person by linearly combining a set of bases using person-specific parameters. Building upon 3DMMs, many works have been proposed to reconstruct and animate human faces by estimating the person-specific parameters given a single-view portrait image [14; 17; 11; 34]. While 3DMMs provide a strong prior for understanding of human faces, they are limited in two ways. First, they exclusively focus on the facial region and fail to capture other characteristic details such as hairstyle, eye glasses, inner mouth *etc*. Second, the geometry and texture fidelity of the reconstructed 3D faces are limited by mesh resolution, leading to unrealistic appearance in the rendered images. In this work, we present a method that effectively exploits the strong prior in 3DMMs while addressing its geometry and texture fidelity limitation by employing neural radiance fields [39; 5; 41].

### 2.2 2D Expression Transfer

The impressive performance of Generative Adversarial Networks (GANs) [22] spurred another line of head avatar animation methods [57; 65; 67; 48; 46; 70; 15]. Instead of reconstructing the underline 3D shape of human faces, these methods represent motion (*i.e*. expression and pose) as a warping field. Expression transfer is carried out by applying a warping operation onto the source image to match the motion of the driving image. By leveraging the powerful capacity of generative models, these methods produce high fidelity results with more realistic appearance compared to 3DMM-based

methods. However, without an explicit understanding and modeling of the underlying 3D geometry of human faces, these methods usually suffer from warping artifacts, unrealistic distortions and undesired identity change when the target pose and expression are significantly different from the ones in the source image. In contrast, we explicitly reconstruct the underline 3D geometry and texture of a portrait image, enabling our method to produce more realistic synthesis even in cases of large pose change during animation.

## 2.3 Neural Head Avatars

Neural Radiance Field (NeRF) [39; 5; 41] debuts remarkable performance for 3D scene reconstruction. Many works [19; 42; 54; 55; 72; 2; 43; 20; 21; 23; 4; 31; 73; 3] attempt to apply NeRF to human portrait reconstruction and animation by extending it from static scenes to dynamic portrait videos. Although these methods demonstrate realistic reconstruction results, they inefficiently learn separate networks for different identities and require thousands of frames from a specific individual for training. Another line of works focus on generating a controllable 3D head avatar from random noise [53; 61; 51; 40; 62; 33; 75; 50]. Intuitively, 3D face reconstruction and animation could be achieved by combining these generative methods with GAN inversion [47; 18; 64; 60]. However, the individual optimization process for each frame during GAN inversion is computationally infeasible for real-time performance in applications such as video conferencing. Meanwhile, several works [56; 66; 6; 13] focus on reconstructing 3D avatars from arbitrary input images, but they cannot animate or reenact these avatars. Closest to our problem setting, few works explore portrait reconstruction and animation in a few-shot [68] or one-shot [24; 29; 38; 76] manner. Specifically, the ROME method [29] combines a learnable neural texture with explicit FLAME meshes [36] to reconstruct a 3D head avatar, encompassing areas beyond the face region. However, using meshes as the 3D shape representation prevents the model from producing high-fidelity geometry and appearance details. Instead of using explicit meshes as 3D representation, the HeadNeRF [24] and MofaNeRF methods learn implicit neural networks that take 3DMM parameters (*i.e.* identity and expression coefficients or albedo and illumination parameters) as inputs to predict the density and color for each queried 3D point. Additionally, the OTAvatar [38] method proposes to disentangle latent style codes from a pre-trained 3D-aware GAN [9] into separate motion and identity codes, enabling facial animation by exchanging the motion codes. Nonetheless, all three models [24; 76; 38] require laborious test-time optimization, and struggle to reconstruct photo-realistic texture details of the given portrait image presumably because they encode the appearance using a compact latent vector. In this paper, we propose the first 3D head neural avatar animation work that not only generalizes to unseen identities without test-time optimization, but also captures intricate details from the given portrait image, surpassing all previous works in quality.

## 3 Method

We present a framework that takes a source image $I_s$ together with a target image $I_t$ as inputs, and synthesizes an image $I_o$ that combines the identity from the source image and the motion (*i.e.*, expression and head pose) from the target image. The overview of the proposed method is illustrated in Fig. 1. Given a source image including a human portrait, we begin by reconstructing the coarse geometry and fine-grained person-specific details via a canonical branch and an appearance branch, respectively. To align this reconstructed 3D neural avatar with the expression in the target image, we employ an off-the-shelf 3DMM [14] model [1] to produce a frontal-view rendering that combines the identity from the source image with the expression from the target image. Our expression branch then takes this frontal-view rendering as input and outputs a tri-plane that aligns the reconstructed 3D avatar to the target expression. By performing volumetric rendering from the target camera view and applying a super-resolution block, we synthesize a high-fidelity image with the desired identity and motion. In the following, we describe the details of each branch in our model, focusing on answering three questions: a) how to reconstruct the coarse shape and texture of a portrait image with neutral expression in Sec. 3.1; b) how to capture appearance details in the source image in Sec. 3.2; and c) how to model and transfer expression from the target image onto the source image in Sec. 3.3. The super-resolution module and the training stages with associated objectives will be discussed in Sec. 3.4 and Sec. 3.5, respectively.

---

[1]We introduce the preliminary of the 3DMM in the supplementary.

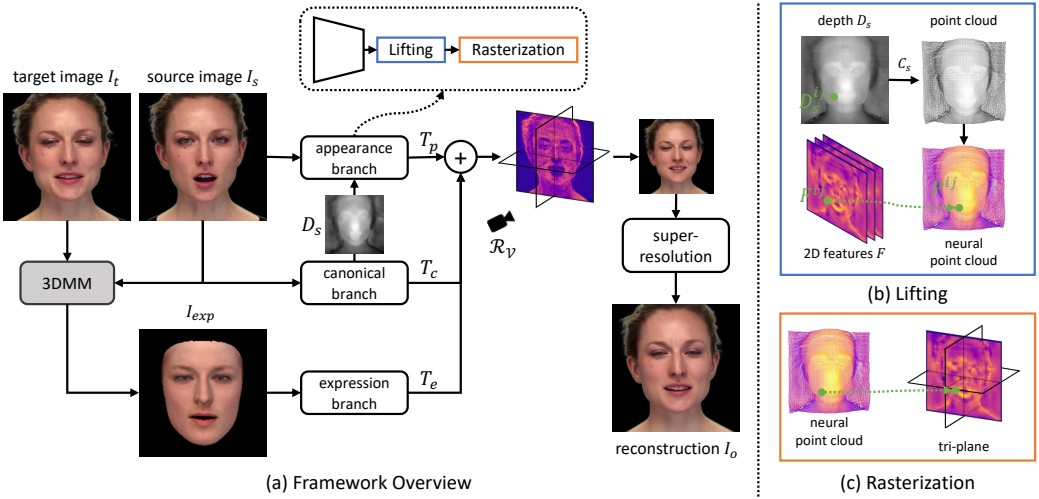

Figure 1: **Overview.** The proposed method contains four main modules: a canonical branch that reconstructs the coarse geometry and texture of a portrait with a neutral expression (Sec. 3.1), an appearance branch that captures fine-grained person-specific details (Sec. 3.2), an expression branch that modifies the reconstruction to desired expression, and a super-resolution block that renders high-fidelity synthesis (Sec. 3.4).

## 3.1 Coarse Reconstruction via the Canonical Branch

Given a source image $I_s$ depicting a human portrait captured from the camera view $C_s$, the canonical branch predicts its coarse 3D reconstruction represented as a tri-plane [9; 10] $T_c$. To serve as a strong geometric prior for the subsequent detailed appearance and expression modeling, we impose two crucial properties on the coarse reconstruction. First, the coarse reconstruction of face images captured from different camera views should be aligned in the 3D canonical space, allowing the model to generalize to single-view portrait images captured from arbitrary camera views. Second, we enforce the coarse reconstruction to have a *neutral* expression (*i.e.*, opened eyes and closed mouth), which facilitates the expression branch to add the target expression effectively.

Based on these two goals, we design an encoder $E_c$ that takes the source image $I_s \in \mathbb{R}^{3 \times 512 \times 512}$ as input and predicts a canonicalized tri-plane $T_c \in \mathbb{R}^{3 \times 32 \times 256 \times 256}$. Specifically, we fine-tune a pre-trained SegFormer [59] model as our encoder, whose transformer design enables effective mapping from the 2D input to the canonicalized 3D space. Furthermore, to ensure that $T_c$ has a neutral expression, we employ a 3DMM [14] to render a face with the same identity and camera pose of the source image, but with a neutral expression. We then encourage the rendering of $T_c$ to be close to the 3DMM's rendering within the facial region by computing an L1 loss and a perceptual loss [25; 69] between them:

$$
\begin{aligned}
I_c &= \mathcal{R}_\mathcal{V}(T_c, C_s) \\
I_{neu}, M_{neu} &= \mathcal{R}_\mathcal{M}(\alpha_s, \beta_0, C_s) \\
\mathcal{L}_{neutral} &= ||I_{neu} - I_c \times M_{neu}|| + ||\phi(I_{neu}) - \phi(I_c \times M_{neu})||,
\end{aligned}
\tag{1}
$$

where $\mathcal{R}_\mathcal{V}(T, C)$ is the volumetric rendering of a tri-plane $T$ from the camera view $C$, $\phi$ is a pre-trained VGG-19 network [49]. $I, M = \mathcal{R}_\mathcal{M}(\alpha, \beta, C)$ is the 3DMM [14] that takes identity coefficients $\alpha$, expression coefficients $\beta$ as inputs, and renders an image $I$ and a mask $M$ including only the facial region from camera view $C$. By setting $\alpha = \alpha_s$ (*i.e.* the identity coeffects of $I_s$) and $\beta_0 = \mathbf{0}$ in Eq. 1, we ensure that $I_{neu}$ has the same identity of $I_s$ but with a neutral expression.

As shown in Fig. 2(c), the rendered image $I_c$ from the canonical tri-plane $T_c$ indeed has a neutral expression with opened eyes and closed mouth, but lacks fine-grained appearance. This is because mapping a portrait from the 2D input to the canonicalized 3D space is a challenging and holistic process. As a result, the encoder primarily focuses on aligning inputs from different camera views and neglects individual appearance details. Similar observations have also been noted in [13; 66]. To resolve this issue, we introduce an appearance branch that spatially transfers details from the input image to the learned coarse reconstruction's surface in the next section.

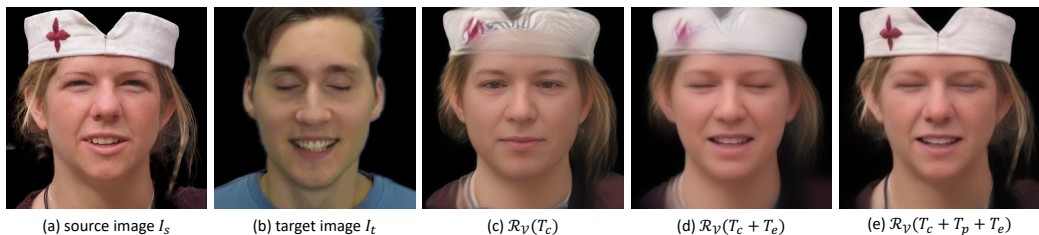

| (a) source image $I_s$ | (b) target image $I_t$ | (c) $\mathcal{R}_\mathcal{V}(T_c)$ | (d) $\mathcal{R}_\mathcal{V}(T_c + T_e)$ | (e) $\mathcal{R}_\mathcal{V}(T_c + T_p + T_e)$ |

Figure 2: **Visualization of the contribution of each branch.** (a) Source image. (b) Target image. (c) Rendering of the canonical tri-plane. (d) Rendering of the combination of the canonical and expression tri-planes. (e) Rendering of the combination of all three tri-planes.

## 3.2 Detail Reconstruction via the Appearance Branch

We now introduce the appearance branch that aims to capture and reconstruct intricate facial details in the input image. The core idea is to leverage the depth map rendered from the canonical tri-plane $T_c$ to compute the 3D position of each pixel in the image such that the facial details can be accurately "transferred" from the 2D input image to the 3D reconstruction. Specifically, we first render $T_c$ from the source camera view $C_s$ to obtain a depth image $D_s \in \mathbb{R}^{128 \times 128}$. The 3D position (denoted as $P^{ij}$) of each pixel $I_s^{ij}$ in the source image $I_s$ can be computed by $P^{ij} = \mathbf{o} + D_s^{ij}\mathbf{d}$, where $\mathbf{o}$ and $\mathbf{d}$ are the ray origin and viewing direction sampled from the camera view $C_s$ of the source image. Based on the 3D locations of all pixels, we construct a neural point cloud [63; 52] by associating the color information from each pixel $I_s^{ij}$ in the 2D image to its corresponding 3D position $P^{ij}$. Instead of directly using the RGB color of each pixel, we employ an encoder $E_p$ to extract 2D features (denoted as $F \in \mathbb{R}^{32 \times 128 \times 128}$) from $I_s$ and associate the feature at each pixel to its corresponding 3D location. As a result, we establish a neural point cloud composed of all visible pixels in the image and associate each point with a 32-dimensional feature vector. This mapping process from a 2D image to the 3D space, is referred to as "Lifting" and demonstrated in Fig. 1(b).

To integrate the neural point cloud into the canonical tri-plane $T_c$, we propose a "Rasterization" process (see Fig. 1(c)) that converts the neural point cloud to another tri-plane denoted as $T_p$ such that it can be directly added to $T_c$. For each location on the planes (*i.e.* the XY-, YZ-, XZ-plane) in $T_p$, we compute its nearest point in the neural point cloud and transfer the feature from the nearest point onto the query location on the plane. A comparison between Fig. 2(d) and Fig. 2(e) reveals the contribution of our appearance tri-plane $T_p$, which effectively transfers the fine-grained details (*e.g.*, pattern on the hat) from the image onto the 3D reconstruction.

## 3.3 Expression Modeling via the Expression Branch

Expression reconstruction and transfer is a challenging task. Naively predicting the expression from an image poses difficulties in disentangling identity, expression, and head rotation. Meanwhile, 3DMMs provide a well-established expression representation that captures common human expressions effectively. However, the compact expression coefficients in 3DMMs are highly correlated with the expression bases and do not include spatially varying deformation details. As a result, conditioning a network solely on these coefficients for expression modeling can be challenging. Instead, we propose a simple expression branch that fully leverages the expression prior in any 3DMM and seamlessly integrates with the other two branches. The core idea is to provide the model with target expression information using a 2D rendering from the 3DMM instead of the expression coefficients. As shown in Fig. 1(a), given the source image $I_s$ and target image $I_t$, we predict their corresponding shape and expression coefficients denoted as $\alpha_s$ and $\beta_t$ respectively using a 3DMM prediction network [14]. By combining $\alpha_s$ and $\beta_t$, we render a *frontal-view* facial image as $I_{exp} = \mathcal{R}_\mathcal{M}(\alpha_s, \beta_t, C_{front})$, where $C_{front}$ is a pre-defined frontal camera pose. We then use an encoder (denoted as $E_e$) that takes $I_{exp}$ as input and produces an expression tri-plane $T_e \in \mathbb{R}^{3 \times 32 \times 256 \times 256}$. We modify the canonical tri-plane $T_c$ to the target expression by directly adding $T_e$ with $T_c$. Note that we always render $I_{exp}$ in the pre-defined frontal view so that the expression encoder can focus on modeling expression changes only and ignore motion changes caused by head rotation. Moreover, our expression encoder also learns to hallucinate realistic inner mouths (*e.g.*, teeth) according to the target expression, as the 3DMM rendering $I_{exp}$ does not model the inner mouth region. Fig. 2(d) visualizes the images rendered by combining the canonical and expression tri-planes, where the target expression from Fig. 2(b) is effectively transferred onto Fig. 2(a) through the expression tri-plane.

### 3.4 The Super-resolution Module

By adding the canonical and appearance tri-planes from a source image, together with the expression tri-plane from a target image, we reconstruct and modify the portrait in the source image to match the target expression. Through volumetric rendering, we can obtain a portrait image at a desired camera view. However, the high memory and computational cost of volumetric rendering prevents the model from synthesizing a high-resolution output. To overcome this challenge, existing works [9; 33; 50; 51] utilize a super-resolution module that takes a low-resolution rendered image or feature map as input and synthesizes a high-resolution result. In this work, we follow this line of works and fine-tune a pre-trained GFPGAN [58; 51] as our super-resolution module [51]. By pre-training on the task of 2D face restoration, GFPGAN learns a strong prior for high-fidelity facial image super-resolution. Additionally, its layer-wise feature-conditioning design prevents the model from deviating from the low-resolution input, thereby mitigating temporal or multi-view inconsistencies, as observed in [51].

### 3.5 Model Training

We utilize a two-stage training schedule to promote multi-view consistent reconstructions, as well as to reduce the overall training time. In the first stage, we train our model without the super-resolution module using a reconstruction objective and the neutral expression loss discussed in Sec. 3.1. Specifically, we compare the rendering of a) the canonical tri-plane (*i.e.*, $I_c = \mathcal{R}_\mathcal{V}(T_c, C_t)$), b) the combination of the canonical and expression tri-planes (*i.e.*, $I_{c+e} = \mathcal{R}_\mathcal{V}(T_c + T_e, C_t)$), and c) the combination of all three tri-planes (*i.e.*, $I_{c+e+p} = \mathcal{R}_\mathcal{V}(T_c + T_e + T_p, C_t)$) with the target image via the L1 and the perceptual losses similarly to Eq. 1:

$$\mathcal{L}_1 = ||I_c - I_t|| + ||I_{c+e} - I_t|| + ||I_{c+e+p} - I_t||,$$
$$\mathcal{L}_p = ||\phi(I_c) - \phi(I_t)|| + ||\phi(I_{c+e}) - \phi(I_t)|| + ||\phi(I_{c+e+p}), \phi(I_t)||. \tag{2}$$

Intuitively, applying supervision to different tri-plane combinations encourages the model to predict meaningful reconstruction in all three branches. To encourage smooth tri-plane reconstruction, we also adopt the TV loss proposed in [9]. The training objective for the first stage is $\mathcal{L}^1 = \lambda_1\mathcal{L}_1 + \lambda_p\mathcal{L}_p + \lambda_{TV}\mathcal{L}_{TV} + \lambda_{neutral}\mathcal{L}_{neutral}$, where $\lambda_x$ is the weight of the corresponding objective.

In the second stage, to encourage multi-view consistency, we only fine-tune the super-resolution module and freeze other parts of the model. We use all the losses in the first stage and a dual-discriminator proposed in [9] that takes the concatenation of the low-resolution rendering and the high-resolution reconstruction as input. Specifically, we use the logistic formulation [22; 27; 58] of the adversarial loss $L_{adv} = \mathbb{E}_{(I_o^l, I_o)}\text{softplus}(D(I_o^l \oplus I_o)))$, where $I_o^l$ is the upsampled version of the low-resolution rendered image and $\oplus$ represents the concatenation operation. The overall training objective of the second stage is $\mathcal{L}^2 = \mathcal{L}^1 + \lambda_{adv}\mathcal{L}_{adv}$.

## 4 Experiments

### 4.1 Datasets

**Training datasets.** We train our model using a single-view image dataset (FFHQ [26]) and two video datasets (CelebV-HQ [74] and RAVDESS [37]). For the single-view images in FFHQ, we carry out the 3D portrait reconstruction task, *i.e.*, the source and target images are exactly the same. For the CelebV-HQ and the RAVDESS datasets, we randomly sample two frames from the same video to formulate a pair of images with the same identity but different motion. Furthermore, we observe that parts of videos in the CelebV-HQ and the RAVDESS datasets are fairly static, leading to a pair of source and target images with similar head poses, impeding the model from learning correct 3D shape of portraits. To enhance the learning of 3D reconstruction, we further employ an off-the-shelf 3D-aware GAN model [9] to synthesize 55,857 pairs of images rendered from two randomly sampled camera views, *i.e.*, the source and target images have the same identity and expression but different views.

**Evaluation datasets.** We evaluate our method and the baselines [29; 24; 38; 65] on the CelebA dataset [32] and the testing split of the HDTF dataset [71], following [65; 38]. Note that our method has never seen any image from these two datasets during training while both StyleHeat[65] and OTAvatar [38] are trained using the training split of the HDTF dataset. Nonetheless, our method generalizes well to all validation datasets and achieves competitive performance, as discussed later.

Table 1: **Comparison on CelebA [32].** [†] Evaluated on a subset of CelebA, as discussed in Sec. 4.4.

| Methods | 3D Portrait Reconstruction | | | | | Cross-Identity Reeanct | | | |
|---|---|---|---|---|---|---|---|---|---|
| | L1↓ | LPIPS↓ | PSNR↑ | SSIM↑ | FID↓ | CSIM↑ | AED↓ | APD↓ | FID↓ |
| ROME [29] | 0.032 | 0.085 | 23.47 | 0.847 | 11.00 | 0.505 | **0.244** | 0.032 | 34.45 |
| Ours | **0.015** | **0.040** | **28.61** | **0.946** | **2.457** | **0.531** | 0.251 | **0.023** | **25.26** |
| HeadNeRF[†] [24] | 0.135 | 0.314 | 13.86 | 0.748 | 65.87 | 0.224 | 0.285 | 0.027 | 117.1 |
| Ours[†] | **0.024** | **0.098** | **25.83** | **0.883** | **9.400** | **0.591** | **0.278** | **0.017** | **22.97** |

**Datasets pre-processing.** We compute the camera poses of images in all training and testing datasets using [14] following [9]. As in previous literature [29; 24; 76], background modeling is out of the scope of this work; we further use an off-the-shelf portrait matting method [28] to remove backgrounds in all training and testing images.

## 4.2 Metrics and Baselines

We evaluate all methods for 3D portrait reconstruction, same-identity and cross-identity reenactment.

**3D portrait reconstruction.** We use all 29,954 high-fidelity images[2] in CelebA [32]. We measure the performance by computing various metrics between the reconstructed images and the input images, including the L1 distance, perceptual similarity metric (LPIPS), peak signal-to-noise ratio (PSNR), structural similarity index (SSIM), and Fréchet inception distance (FID).

**Same-identity reenactment.** We follow [38] and use the testing split of HDTF [71], which includes 37,860 frames in total. We use the first frame of each video as the source image and the rest of the frames as target images. Following the evaluation protocol in [38; 65], we evaluate PSNR, SSIM, cosine similarity of the identity embedding (CSIM) based on [12], average expression distance (AED) and average pose distance (APD) based on [14], average keypoint distance (AKD) based on [8], as well as LPIPS, L1 and FID between the reenacted and ground truth frames.

**Cross-identity reenactment.** We conduct cross-identity reenactment on both the CelebA and HDTF datasets. For CelebA, we split the dataset into 14,977 image pairs and transfer the expression and head pose from one image to the other. As for the HDTF dataset, we follow [38] and use one clip as the driving video and the first frame of the other videos as source images, which produces 67,203 synthesized images in total. To fully evaluate the performance of *one-shot* avatar animation, we transfer motion from the HDTF videos to the single-view images in the CelebA dataset. Similar to [65], we use the first 100 frames in the HDTF videos as target images and 60 images sampled from the CelebA as source images, resulting in a total of 114,000 synthesized images. Since there is no ground truth for cross-identity reenactment, we evaluate the results based on the CSIM, AED, APD, and FID metrics. More evaluation details can be found in the supplementary.

**Baselines.** In terms of baselines, we compare our method against a SOTA 2D talking head synthesis method [65], two SOTA 3D head avatar animation methods [24; 29], and one concurrent work [38].

## 4.3 Implementation Details

We implement the proposed method using the PyTorch framework [45] and train it with 8 32GB V100 GPUs. The first training stage takes 6 days, consisting of 750000 iterations. The second training stage takes 2 days with 75000 iterations. Both stages use a batch size of 8 with the Adam optimizer [30] and a learning rate of 0.0001. More implementation details can be found in the supplementary.

## 4.4 Qualitative and Quantitative Results

**3D portrait reconstruction.** For each testing portrait image, we reconstruct its 3D head avatar and render it from the observed view using different methods. By comparing the rendering with the input image, we assess the fidelity of 3D reconstruction of each method. Table 1 shows the quantitative results, demonstrating that our model achieves significantly better reconstruction and fidelity scores. These results highlight the ability of our model to faithfully capture details in the input images and reconstruct high-fidelity 3D head avatars. Visual examples are present in the supplementary.

---

[2]Due to the time-consuming nature of HeadNeRF, we compare our method with HeadNeRF on a subset of 3000 images from the CelebA dataset.

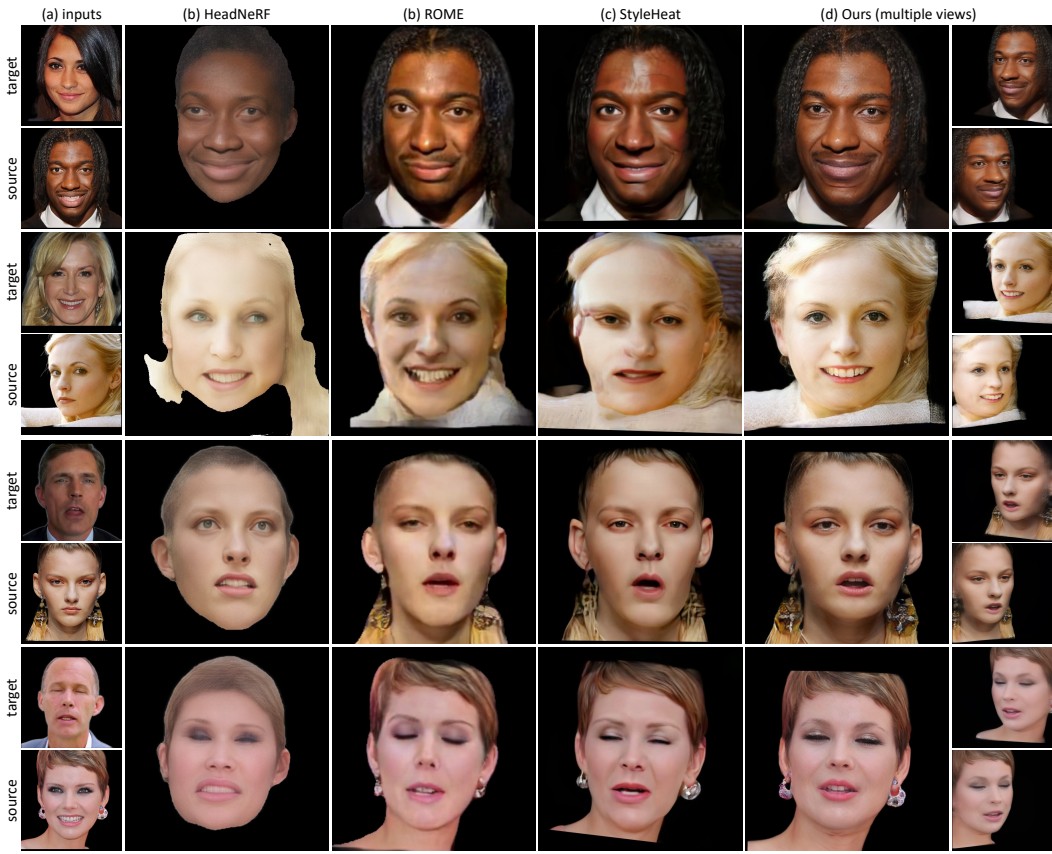

Figure 3: **Cross-identity reenactment on CelebA [32] and HDFT [71].** The first two rows show cross-identity reenactment results on the CelebA dataset, while the last two rows demonstrate motion transfer from videos in the HDFT dataset to images in the CelebA dataset.

Table 2: **Comparison on the HDTF dataset [71].**

| Methods | Same-Identity Reenactment | | | | | | | | | Cross-Identity Reenactment | | | |
| | PSNR↑ | SSIM↑ | CSIM↑ | AED↓ | APD↓ | AKD↓ | LPIPS↓ | L1↓ | FID↓ | CSIM↑ | AED↓ | APD↓ | FID↓ |
| --- | --- | --- | --- | --- | --- | --- | --- | --- | --- | --- | --- | --- | --- |
| ROME [29] | 20.75 | 0.838 | 0.746 | **0.123** | 0.012 | 2.938 | 0.173 | 0.047 | 31.55 | 0.629 | 0.247 | 0.020 | **43.38** |
| OTAvatar [38] | 20.12 | 0.806 | 0.619 | 0.162 | 0.017 | 2.933 | 0.198 | 0.053 | 36.63 | 0.514 | 0.282 | 0.028 | 44.86 |
| StyleHeat [65] | 19.18 | 0.805 | 0.654 | 0.141 | 0.021 | 2.843 | 0.194 | 0.056 | 108.3 | 0.537 | **0.246** | 0.025 | 105.1 |
| Ours | **22.15** | **0.868** | **0.789** | 0.129 | **0.010** | **2.596** | **0.117** | **0.037** | **21.60** | **0.643** | 0.263 | **0.018** | 47.39 |

**Cross-identity reenactment.** Fig. 3, and Fig. 4 showcase the qualitative results of cross-identity reenactment on the CelebA [32] and HDFT [71] dataset. Compared to the baselines [29; 65; 38], our method faithfully reconstructs intricate details such as hairstyles,

Table 3: Cross-identity reenactment between the HDTF dataset [71] and the CelebA dataset [32].

| Methods | CSIM↑ | AED↓ | APD↓ | FID↓ |
| --- | --- | --- | --- | --- |
| ROME [29] | 0.521 | 0.270 | 0.022 | 76.03 |
| StyleHeat [65] | 0.461 | **0.270** | 0.038 | 94.28 |
| Ours | **0.551** | 0.274 | **0.017** | **59.48** |

earrings, eye glasses *etc.* in the input portrait images. Moreover, our method successfully synthesizes realistic appearance change corresponding to the target motion. For instance, our model is able to synthesize plausible teeth when the mouth transitions from closed to open (*e.g.*, row 3 in Fig. 3), it also hallucinates the occluded face region when the input image is in profile view (*e.g.*, row 2 in Fig. 3). In contrast, the mesh-based baseline [29] can neither capture photo-realistic details nor hallucinate plausible inner mouths, while the 2D talking head synthesis baseline [65] produces unrealistic warping artifacts when the input portrait is in the profile view (*e.g.* row 2 in Fig. 3). We provide quantitative evaluations of the cross-identity reenactment results in Table 1, Table 2, and Table 3. Our method demonstrates better fidelity and identity preservation scores, showing its strong ability in realistic portrait synthesis. It is worth noting that HDTF [71] includes images that are less sharp compared to the high-fidelity images our model is trained on, which may account for the slightly lower FID score in Table 2.

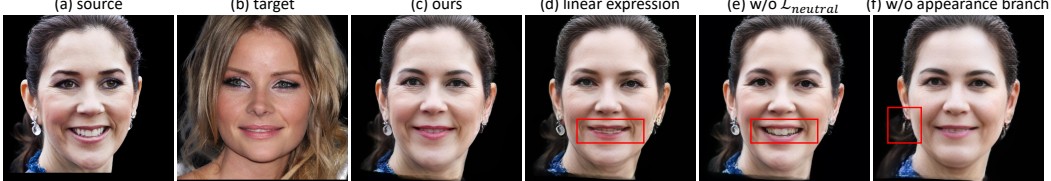

| | (a) source | (b) target | (c) ours | (d) linear expression | (e) w/o $\mathcal{L}_{neutral}$ | (f) w/o appearance branch |

Figure 5: **Ablation studies.** Details are explained in Sec. 4.5.

Table 4: **Ablation studies.** Blue text highlights the inferior performance of the variants. (Sec. 4.5)

| | 3D Portrait Reconstruction | | | | | Cross-Identity Reeanct | | | |
| Methods | L1↓ | LPIPS↓ | PSNR↑ | SSIM↑ | FID↓ | CSIM↑ | AED↓ | APD↓ | FID↓ |
|---|---|---|---|---|---|---|---|---|---|
| w/o perspective | 0.061 | 0.239 | 19.53 | 0.712 | 47.84 | 0.370 | 0.243 | 0.015 | 43.33 |
| w/o neutral constraint | 0.027 | 0.124 | 25.65 | 0.854 | 13.56 | 0.593 | 0.315 | 0.018 | 22.92 |
| linear expression | 0.031 | 0.134 | 25.08 | 0.841 | 14.62 | 0.443 | 0.217 | 0.016 | 26.18 |
| Ours | 0.030 | 0.116 | 24.77 | 0.861 | 10.47 | 0.599 | 0.276 | 0.017 | 17.36 |

**Same-identity reenactment.** Table 2 shows the quantitative results of same-identity reenactment on HDTF [71]. Our method generalizes well to HDTF and achieves better metrics compared to existing SOTA methods [29; 38; 65]. The qualitative results of same-identity reenactment can be found in the supplementary.

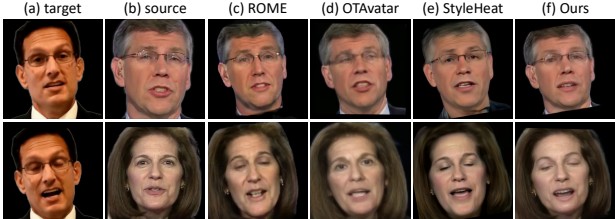

| (a) target | (b) source | (c) ROME | (d) OTAvatar | (e) StyleHeat | (f) Ours |

Figure 4: **Cross-identity reenactment on HDFT [71].**

**Efficiency.** Since HeadNeRF [24] and OTAvatar [38] require latent code optimization for unseen identities, the process of reconstructing and animating an avatar takes them 53.0s and 19.4s, respectively. Meanwhile, ROME [29] and our method only needs an efficient forward pass of the network for unseen identities, taking 1.2s and 0.6s respectively. Overall, our method strikes the best balance in terms of speed and quality.

## 4.5 Ablation Studies

We conduct experiments to validate the effectiveness of the neutral expression constraint (Sec. 3.1), the contribution of the appearance branch (Sec. 3.2), and the design of the expression branch (Sec. 3.3).

**Neutral expression constraint.** In our model, we aim to wipe out the expression in the source image by enforcing the coarse reconstruction from the canonical branch to have a neutral expression. This ensures that the expression branch always animates a fully "neutralized" expressionless face from the canonical branch. Without this, the expression branch fails to correctly modify the coarse reconstruction into the target expression, as shown in Fig. 5(e) and Table 4 (*i.e.*, worse AED score).

**Appearance branch.** The appearance branch is the key to reconstructing intricate facial details of the input portrait image. Without this branch, the model struggles to capture photo-realistic details, resulting in considerably lower reconstruction and fidelity metrics, as shown in Table 4 and Fig. 5(f).

**Alternative expression branch design.** Instead of using the frontal view rendering from the 3DMM to provide target expression information to the expression branch (see Sec. 3.3), an alternative way is to use the 3DMM expression coefficients to linearly combine a set of learnable expression bases. However, this design performs sub-optimally as it overlooks the individual local deformation details caused by expression changes, introducing artifacts (*e.g.*, mouth not fully closed) shown in Fig. 5(d) and lower FID in Table 4. We provide more results and ablations in the supplementary.

## 5 Conclusions

In this paper, we propose a framework for one-shot 3D human avatar reconstruction and animation from a single-view image. Our method excels at capturing photo-realistic details in the input portrait image, while simultaneously generalizing to unseen images without the need for test-time optimization. Through comprehensive experiments and evaluations on validation datasets, we demonstrate that the proposed approach achieves favorable performance against state-of-the-art baselines on head avatar reconstruction and animation.

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
