# OpenReview forum: "Generalizable One-shot 3D Neural Head Avatar"
_NeurIPS.cc/2023/Conference — NeurIPS 2023 poster_

### Official Review · Reviewer_k8VD · 2023-07-01

**Soundness:** 3 good
**Presentation:** 3 good
**Contribution:** 3 good
**Rating:** 6
**Confidence:** 5

**Summary:**

Authors proposed a method for one-shot facial animation with high level of details saved from the input image. To handle facial deformations they disentangle expression and appearance and introduces a neural point cloud renderer for that. To train a model authors used quite popular set of loss functions and data, as well as incorporated synthetic data to improve underlying geometry. Qualitative and quantitative experiments show the advantage of architectural design and superiority over existing methods.


**Strengths:**

- Idea to decompose geometry and appearance in canonical face is an interesting combination of recent success in neural rendering
- The demonstrated method outputs clearly emphasise huge number of saved details and right disentanglement for identity and expression.
- The results contains enormous level of the details compared to the demonstrated baselines.
- Authors carefully describe each branch contributed in the resulted quality.


**Weaknesses:**

The comparison doesn’t have a latent avatar models [1, 2]. It can improve the overall experiments section since both methods produce high-resolutional results for reenactments

- [1] One-Shot Free-View Neural Talking-Head Synthesis for Video Conferencing
- [2] Megaportraits: One-shot megapixel neural head avatars.

The introduced neural point cloud can not cover huge view changes.
Some statements are overconfident with respect to the evaluation (mostly regarding to consistent usage SOTA).
Lack of ethical section in the main text, which is important for human avatars.

Since the emotions at the video are too unrealistic, I want to emphasise that comparison with the methods above is necessary to get more fair outcomes from the paper. The method looks bonded by the pre-aligned geometry and the possibility to express human level poses.
Some minor details:
Writing quality can be improved.
Lack of limitation section.


**Questions:**

The neural point cloud introduced in the line 182 is equivalent to dense feature grid, isn’t it? This procedure also similar to latents avatar like [1, 2], but with orthographic projection.
Could you elaborate the reasons to include synthetic images and how does affect final quality? For instance you mentioned that CelebV-HQ is static (which is debatable) is it a help somehow?
Could you compare expression transfer? As I see in the first video results there is a clear gap of the emotional level or naturally during the reenactment.
What is a time needed for an inference?

**Limitations:**

The method by itself has limited geometry ability to describe human heads. Based on the presented image there is not information about upper body.
There is not option to render 360 degree view paths.
Lack of the realistic emotions.
I doubt that the method can be easily used in real-time applications.

---

> ### Author Rebuttal · Authors · 2023-08-09
>
> **Comparison to latent 2D avatar methods.**
> - Please see A1 in the Global Response above. Note that we cannot compare to Megaportraits since its code is not publicly available. For "One-Shot Free-View Neural Talking-Head Synthesis for Video Conferencing", we used a public third-party implementation because the official code is not released (we have sent the code link of this implementation to AC since we cannot add external links in the rebuttal text).
>
> **Point clouds cannot cover huge view changes.**
> - The neural point clouds indeed only cover the visible parts in the input portrait, but the canonical branch is developed to provide cues of missing parts and guarantee a realistic full reconstruction. This is exactly our main motivation for disentangling coarse geometry and intricate details into two tri-planes. The former learns to map a portrait image in any pose to a canonical coarse reconstruction implicitly in a holistic manner, while the latter explicitly lifts the partial observation from the 2D space using camera pose and rendered depth. Through training, the two branches learn to be compatible with each other and produce reasonable full reconstructions for inputs captured from various views. As shown in Fig.1 (right) in the rebuttal PDF, even for a source image captured from an extreme profile view, our method produces reasonable reconstruction, while 2D baselines present unrealistic warping artifacts. We do think that further improvement is possible for the neural point cloud part. For instance, one way is to inpaint the neural point cloud by utilizing the symmetry prior of human portraits, we leave this exploration in future works.
>
> **Synthetic images.**
> - Why do we need synthetic data? We first analyze the variation between source and target poses in the CelebV-HQ and the synthetic datasets. We randomly sample 1000 training pairs and compute the pose distance between the source and target image in each pair. The average pose difference in CelebV-HQ and the synthetic dataset is 0.0482 and 0.0985, respectively. This shows that sampled pairs in CelebV-HQ have less pose variation. Instead, the source and target images in the synthetic data have larger pose variation, which enforces the model to hallucinate realistic missing parts during training. We further carry out an ablation study that learns a model without using synthetic data. As shown in the last column in Fig.4 in the rebuttal PDF,  the model learned without synthetic data shows artifacts in the occluded region (e.g., the left side of the nose).
> - What alternative generative models could be used to produce synthetic data and how does it affect our model? We kindly point to A2 in the Global Response above, where we show that by replacing EG3D with Next3D [2] in the training data synthesis process, our model can be further improved.
> - We would like to emphasize that utilizing pre-trained generative models for data synthesis demands minimal effort. In our case, the synthesis of over 50,000 training images using either EG3D or Next3D takes just a few hours.
>
> [2] Next3D: Generative Neural Texture Rasterization for 3D-Aware Head Avatars
>
> **Inference time.**
> - We have discussed efficiency in Line 333 - 337 in the main paper. We emphasize that our method is efficient since it is generalizable and does not require any person-specific optimization or test-time fine-tuning. Real-time is not a primary focus in this paper. However, we do identify possible acceleration methods such as replacing SegFormer encoders with more light-weight networks, reducing rendering resolution, or using mixed precision inference.
>
> **Upper body and 360 degree view paths.**
> - Our method outperforms existing methods by capturing high fidelity details in regions beyond the face area (e.g., hair, earrings). Upper body human capture is related to human body deformation and especially challenging in the generalizable setting. Even in works that overfit a model to a specific identity [3], reconstructing the upper body is non-trivial and requires a dedicated neural radiance field. We leave upper body reconstruction for future research. Similarly, 360 degree head reconstruction requires advanced representations than the native tri-plane and special training data, as discussed in [4]. Though we believe that our method can generalize to full heads using the new techniques and data in [4], it is not the focus of this paper and is left out for future research.
>
> [3] AD-NeRF: Audio Driven Neural Radiance Fields for Talking Head Synthesis. ICCV 2021.
>
> [4] PanoHead: Geometry-Aware 3D Full-Head Synthesis in 360deg. CVPR. 2023.
>
> **Ethical and limitation section.**
> - We have discussed the limitations and the social impact, in detail, in the supplementary document and will move them to the main paper in the next revision.
>
> **Overconfident statements.**
> - We will remove SOTAs.
>
> **Limited geometry ability.**
> - Please see A2 in the Global Response, Fig.4 in the rebuttal PDF, and the video we sent to AC.  We show the extracted mesh in Fig.4 in the rebuttal PDF, demonstrating that the expression branch can deform the underline geometry of the canonical tri-plane to match extreme expressions (e.g., wide open mouth) so long as we use balanced training data.

---

> > ### Comment · Reviewer_k8VD · 2023-08-21
> >
> > Following a careful review of their response and the input from other reviewers (especially in conversation with R. PU1D), I am convinced that the additional experiments they carried out have led to a more comprehensive evaluation of their proposed approach. This has undoubtedly supported the quality of the work, and it's worth noting that the authors gave clear explanations on all aspects. I will keep my original rating (WA) and would tend for acceptance.

---

### Official Review · Reviewer_PU1D · 2023-07-04

**Soundness:** 2 fair
**Presentation:** 3 good
**Contribution:** 2 fair
**Rating:** 4
**Confidence:** 4

**Summary:**

The method aims to build a generalizable model to create an animatable 3D human head representation from a single-view portrait source image. The resulting representation can be used to reenact the source image with target images with different subjects and expressions.

The key idea is to use three tri-planar representations (T_c, T_p, and T_e) with the underlying 3DMM from [14].

T_c is a canonical tri-plane representing the neutral source face. Since the source face can be in arbitrary expressions, this branch is responsible to undo the expression. An encoder pretrained with SegFormer [59] maps the input source image to a tri-plane with 32 feature channels. To enforce T_c to be in the neutral face, this branch relies on [14] to extract the identity and expression coefficients. The neutral expression obtained by zeroing out the expression coefficients

T_p is responsible for capturing detailed appearance. Neural point cloud representation is constructed from the depth image of T_c viewed from the camera of the source image, with features associated with the source image, again obtained from an encoder pretrained with SegFormer [59]. This neural point cloud is finally rasterized to the appearance tri-plane T_p.

T_e represents a tri-plane based on the 3DMM of the source identity and the target expression extracted with [14]. The frontal view rendering of this identity and expression is passed to an encoder pretrained with SegFormer [59] to obtain an expression tri-plane T_e.

The image generated from the sum of three tri-planes goes through a super-resolution with a pre-trained GFPGAN to obtain the final image.

The method is compared against ROME [29] and StyleHeat [65] in the main paper. It is also compared against a recent work Next3D in the supplementary.

**Strengths:**

The use of the different tri-planar representations, each making use of the information from the 3DMM prediction, is interesting.

A setup only requiring a feedforward pass without test-time optimization is simple and practical.

**Weaknesses:**

A naive addition of three tri-plane representations may not make sense. T_c is representing the neutral face while T_p is obtained from the source image with an expression. A corresponding position on a face will have the spatial coordinates in T_c and T_p, making this addition insensible. Perhaps consider warping the tri-planes before adding them?

The first sentence in 3.2 says the appearance branch is to reconstruct intricate facial details. This may be somewhat misleading. What the appearance branch is really responsible for is capturing the details of the non-facial regions. This is clear from Fig. 2 (e) showing a better reconstruction of the hat, hair, and necklace. Fig. 5 (f) is also highlighting the earring.

Given the above, the proposed setup would have made more sense if the appearance branch masked out the facial region, in contrast to the canonical branch masking out the non-facial region when taking the loss.

Perhaps related to this, the video shows visual artifacts. The result looks like the facial parts are animated textures on a non-deforming geometry. This is clear when the target subject opens the jaw at 0:20 and 1:04 but the resulting face opens the mouth without moving the jaw. This is a sign that the neutral face geometry is having too much influence.

**Questions:**

If authors have more thoughts on why the visual artifact with the jaw open and extreme expressions happens, I would like to hear them. Other than the naive composition of the tri-planes, I suspect the underlying 3DMM may not be sufficient to capture realistic deformations.

**Limitations:**

As mentioned in the weakness, the method exhibits clear artifacts with large facial deformations. This should be clarified.

---

> ### Author Rebuttal · Authors · 2023-08-09
>
> **Appearance branch.**
> - Please see A3 in the Global Response above. As discussed in the supplementary (Line 105 - 108), to prevent the source expression from leaking into the final animation results, the appearance branch masks out the facial region in the input source image and only captures the non-facial details, while the expression reconstruction solely coming from the expression branch. We will clarify this more in the next revision.
>
> **Performance and artifacts in rare expressions.**
> - The proposed method efficiently generalizes a single network to unseen identities. while demonstrating competitive 3D head avatar animation performance quantitatively and qualitatively against multiple state-of-the-art baselines. As shown in Sec.4 in the main paper and Sec.3 in the supplementary, our model can produce realistic animation results for most common target expressions.
> - For the "jaw opening artifacts", please see A2 in the Global Response, Fig.4 in the rebuttal PDF, and the video we sent to AC. We emphasize that the mouth-wide-open expression falls into the long-tail distribution of the training dataset, causing the artifacts. It does not demonstrate a fundamental flaw in our framework's design. With proper training data discussed in A2 above, our method can naturally deform the jaw when the mouth is wide open. We also show the extracted mesh in Fig.4 in the rebuttal PDF, demonstrating that the expression branch indeed changes not only the texture but also the underline geometry of the canonical tri-plane.

---

> > ### Comment · Reviewer_PU1D · 2023-08-15
> > **Not capturing jaw open, a very common facial deformation, is a flaw in design**
> >
> > The two frames I pointed out where the artifacts of the underlying geometry not deforming are just examples to clarify the issue. Overall the animations look creepy because the texture is unnaturally swimming over the facial geometry with little deformations. I argue that jaw opening is a very important part of facial deformation which any facial system should strive to do well at. Extreme jaw openings may appear less frequently in the data, but the inability to capture such deformations is a sign that the model cannot handle more subtle jaw openings appearing more frequently. An objective proof of the importance of the jaw opening is how the FLAME facial model decided to put an explicit LBS-based jaw control on top of the PCA-based facial deformations.
> >
> > As I noted in my initial review, the setup would have made more sense if the T_c representing the neutral face was warped according to the underlying 3DMM, and then added together with other triplanes. It is not a surprise to me that T_c without warping failed to capture facial deformations.

---

> > > ### Author Response · Authors · 2023-08-15
> > > **The "Jaw-opening issue" can be resolved when we use balanced training data**
> > >
> > > We thank the reviewer for the response.
> > >
> > > We agree that “naturally modeling jaw opening” is of crucial importance. We kindly point the reviewer to *Fig.4 in the rebuttal PDF* and *A2 in the Global Response above*, where the *“jaw opening” issue is resolved* by using balanced training data with more jaw-opening images. This validates our assumption that the “jaw opening” issue is caused by the long-tail training data, rather than a fundamental design flaw in our framework.
> > >
> > > For the warping idea, we kindly point the reviewer to *A3 in the Global Response above*. We have tried applying explicitly warping such as a learnable flow or a TPS transformation to the appearance tri-plane but found all led to trivial solutions (i.e., the tri-plane collapses to a thin plane). Interestingly, we found an effective implicit warping design and has discussed it in details in *A3 in the General Response* above.
> > >
> > > Please let us know if you have more questions.

---

> > > > ### Comment · Reviewer_PU1D · 2023-08-15
> > > > **Video?**
> > > >
> > > > How can I check the video sent to the AC? I am especially interested in inspecting the deformation details.

---

> > > > > ### Comment · Reviewer_PU1D · 2023-08-18
> > > > > **Isn't Next3D a concurrent competing method? Using a competing method for training does not make sense**
> > > > >
> > > > > I checked the video. Yes, the deformation looks far better. But is this not just proving that Next3D is able to model facial deformations better?
> > > > >
> > > > > * This paper trained with EG3D (as in L257): flaw in facial deformations, e.g. jaw open
> > > > > * Next3D trained with EG3D: good facial deformations
> > > > > * This paper trained with Next3D: good facial deformations (obvious - does not justify the proposed design)
> > > > >
> > > > > To be fair, this paper should be reviewed only with the version trained with EG3D.

---

> > > > > > ### Author Response · Authors · 2023-08-18
> > > > > > **Next3D is a generative model, it cannot be used for portrait reconstruction and animation without PTI**
> > > > > >
> > > > > > Thanks for checking out the video. We are glad that the reviewer thinks the “deformation looks far better” in the video.
> > > > > >
> > > > > > Here are some clarifications about Next3D:
> > > > > >
> > > > > > - Our method focuses on portrait reconstruction and animation, while Next3D [1]  is a *3D generative adversarial network* like EG3D [2]. The underlying motivation, goals and formulation of our method and these two generative models are fundamentally different. *As a generative model, Next3D aims to synthesize 3D portrait of random identity, It cannot be used for portrait reconstruction and animation given a single-view image in its native formulation.*
> > > > > > - When combined with PTI [3], Next3D can be used for portrait reconstruction and animation. This is why the baseline in the supplementary (Sec. 2.1) is called "Next3D-PTI" rather than "Next3D". As shown qualitatively and quantitatively, the performance of this "Next3D-PTI" baseline is inferior to our method. Besides, the PTI process alone requires 5 minutes, while our method only takes 0.6 seconds for the entire reconstruction process. *Besides Next3D-PTI, our method trained with EG3D data already outperformed 12 other baselines quantitatively in the main paper and this rebuttal.*
> > > > > > - Next3D is trained with single-view images (i.e., FFHQ), we never show or mention anywhere about training Next3D using data synthesized by EG3D.
> > > > > > - When we use Next3D to synthesize training data, we use it as a generative model in its native formulation (i.e., without PTI).
> > > > > >
> > > > > > The key argument here is “whether the jaw opening artifacts are caused by a fundamental design flaw in our model?”. We respectfully argue that it is not a fundamental design flaw by comparing two models:
> > > > > >
> > > > > > - Our model trained with a biased dataset (i.e., synthesized by EG3D), which shows natural deformation for most human expressions, but has artifacts for the “jaw opening” expression that is rare in the training dataset.
> > > > > > - Our model trained with a balanced dataset (i.e., synthesized by Next3D), which shows natural deformation for the “jaw opening” expression and other expressions.
> > > > > >
> > > > > > We emphasize that in these two experiments, the architecture and training hyper-parameters are the same. The only difference is the training data. Thus, we argue that the “jaw opening issue” is caused by a biased dataset, instead of a design flaw in our model.
> > > > > >
> > > > > > We encourage the reviewer to check the related works listed below and Sec.2.1 in the supplementary for more details. Please let us know if you have further questions.
> > > > > >
> > > > > > [1] Next3D: Generative Neural Texture Rasterization for 3D-Aware Head Avatars. CVPR 2023.
> > > > > >
> > > > > > [2] Efficient Geometry-aware 3D Generative Adversarial Networks. CVPR 2022.
> > > > > >
> > > > > > [3] Pivotal Tuning for Latent-based editing of Real Images. ACM TOG 2022.

---

> > > > > > > ### Comment · Reviewer_PU1D · 2023-08-20
> > > > > > >
> > > > > > > I appreciate the authors for their thorough clarifications. I agree with the authors that Next3D is not exactly a competing method, though it can do reenactments on arbitrary target faces after a time-consuming latent space optimization.
> > > > > > >
> > > > > > > I still want to say the following.
> > > > > > >
> > > > > > > * If the paper is to be reviewed with the model trained with Next3D, it must redo all the evaluations, especially the ablations. With more powerful data from Next3D, it is very possible that a much simpler setup is sufficient to achieve the goal.
> > > > > > >
> > > > > > > * The authors say the deformation artifact is due to the lack of data. I say that the value of many graphics applications is about dealing with data scarcity through model-based approaches, e.g. 3DMM. The proposed approach, while introducing a complex setup with multiple tri-planar representations and 3DMM, failed to capture important facial deformations if not brute-forced with data.
> > > > > > >
> > > > > > > To pay respect to the discussions and clarifications, I bumped the score slightly.

---

> > > > > > > > ### Author Response · Authors · 2023-08-20
> > > > > > > > **Reply to the comments**
> > > > > > > >
> > > > > > > > We thank the reviewer for providing additional comments and for updating the score. We appreciate the ongoing discussion and your valuable insights.
> > > > > > > >
> > > > > > > > We would like to take this opportunity to highlight the following points:
> > > > > > > > - Our main contribution is a generalizable and high-fidelity portrait animation and reconstruction model. This model overcomes the limitation of both implicit methods (which struggle to generalize to unseen identity) and 3DMM methods (which cannot capture details beyond facial region). The model in the submission (i.e., without Next3D data) sets a new benchmark in this field by quantitatively outperforming a dozen state-of-the-art methods.
> > > > > > > > - We agree that designing models that leverage limited data is important. But it is not the main focus of this paper. The experiments that utilize data synthesized by Next3D serve as a means to demonstrate that the observed "jaw-opening issue" is not rooted in a fundamental design flaw.
> > > > > > > > - Utilizing pre-trained generative models for data synthesis demands minimal effort. In our case, the synthesis of over 50,000 training images using either EG3D or Next3D takes just a few hours.
> > > > > > > > - We will include comprehensive ablation studies about the effectiveness of training data in the next revision.
> > > > > > > >
> > > > > > > > Please let us know if you have any further questions.

---

### Official Review · Reviewer_8oNf · 2023-07-04

**Soundness:** 3 good
**Presentation:** 4 excellent
**Contribution:** 2 fair
**Rating:** 6
**Confidence:** 3

**Summary:**

The paper proposes a novel approach for building single-shot animatable head avatars. This is achieved through three branches: the canonical reconstruction branch, the detailed appearance branch, and the expression branch, all of which are represented as tri-plane nerfs. The three tri-planes are added together to form the final 3D representation. The paper also features a super-resolution module, which not only achieves much better photorealism than existing methods but also nicely hallucinates teeth while retaining multi-view consistency.



**Strengths:**

The usage of separate tri-planes for canonical, appearance details, and expressions is a novel approach (although there are some weaknesses associated with this design choice, as discussed in the following sections).
Along with the super-resolution block, the method generates high-quality images and surpasses previous works in various image quality and diversity metrics. The proposed expression branch can successfully hallucinate teeth during deformation. Additionally, multi-view consistency is well-preserved, as demonstrated in the supplementary video. Overall, the approach is thoroughly evaluated and compared with state-of-the-art (SOTA) baselines.



**Weaknesses:**

Weakness of the result:
The method has difficulty with jaw opening. In the supplementary video, when the person opens the mouth, only the lips move and not the jaw. I suspect that this is because the additive expression branch is not powerful enough to model such big deformation. What’s the author’s explanation of this?

Appearance branch:
Using the depth of the source image, the paper proposes to back-project the source image feature to 3D space in the appearance branch. It seems to me that this branch not only adds canonical appearance details but also would entangle the source expressions into the 3D reconstruction. Since the expression branch does not take into account the source expression parameters, how does the method remove the source expression from the 3D reconstruction? Does the network learn to ignore expression-dependent details in the appearance branch through neural network magic? How does R_v(T_c + T_p) look like in Figure 2?

Invalid ablation on the expression branch:
The paper shows that the proposed expression branch performs better compared to a ‘linear expression’ ablation. However, this ablation baseline does not make sense to me because the 3DMM expressions are linear blendshapes, i.e. the warping fields are linearly combined, not the 3D volumes. Therefore, expressions cannot be expressed as the weighted average of expression triplanes. This ablation setup is not valid in my opinion.


**Questions:**

Please see the first two points of the weaknesses section.


**Limitations:**

Yes, the author adequately discussed limitation and negative societal impact in the supplementary material.

---

> ### Author Rebuttal · Authors · 2023-08-09
>
> **Weakness of the result.**
> - Please see A2 in the Global Response above, Fig.4 in the PDF and the video we sent to AC. The "jaw opening" issue is caused by the long-tail expression distribution of the training dataset rather than using the additive expression branch. With training data containing a more balanced distribution of large facial expressions, our method naturally deforms the jaw when the mouth is wide open.
>
> **Appearance branch.**
> - Please see A3 in the Global Response above, the visualization of $R_v(T_c + T_p)$ can be found in Fig.2 (a) in the rebuttal PDF. In the submission, we remove the source expression by masking out the expression-dependent regions (i.e., eyes and mouth) before feeding the source image into the appearance branch. In A3 in the Global Response above, we further show that learning a model that implicitly ignores the source expression is feasible and shows promising preliminary results.
>
> **Invalid ablation on the expression branch.**
> - We will remove this ablation study in the next revision.

---

> > ### Comment · Reviewer_8oNf · 2023-08-22
> >
> > The rebuttal partially resolved my concerns and I will keep my original score.

---

### Official Review · Reviewer_FcT2 · 2023-07-06

**Soundness:** 3 good
**Presentation:** 3 good
**Contribution:** 3 good
**Rating:** 6
**Confidence:** 4

**Summary:**

This paper presents a method for generating Neural Head Avatars given a single image of a subject. More specifically, the input is a source image of a person as well as a target image specifying the expression, and the goal is to generate a rendering of the person in the source image with the expression of the target image. To do so, the method predicts 3 sets of tri-planes modelling the coarse geometry and appearance of the source image, and the expression of the second image. The geometry and appearance are modeled in a canonicalized 3D pose to encourage robustness to different poses and varying viewpoints. After training, the method can be applied to unseen images and generate animatable avatars without requiring any expensive test-time optimization. Overall the experiments validate the effectiveness of the approach.

**Strengths:**

1. The methodology of the paper is technically sound. The components of the method are sufficiently motivated and their usefulness is assessed in the ablation study.
2. The paper has very good experimental results. Quantitatively, it performs favorably compared to the state-of-the-art in the CelebA and HDTF datasets. The qualitative results are also very good.
3. The paper is well-written and easy to follow. Most components are adequately described in the main text and the figures are relatively easy to parse (with the exception of some parts of Figure 1. where the flow of information is not immediately clear).

**Weaknesses:**

The paper does not have any particularly important weak points. Occasionally there are some artifacts in the reconstruction for parts of the face that are not visible in the source image (e.g. Row 2 of Figure 3).

**Questions:**

1. I was wondering how accurate is the 3DMM prediction stage, and how much this affects the overall quality of the reconstructions.

**Limitations:**

The paper fails to discuss the potential societal impacts of this work. One obvious misuse of the technology could be in DeepFakes. I believe that the authors should include a more detailed overview of the potential misuses and what safeguards can be used.

---

> ### Author Rebuttal · Authors · 2023-08-09
>
> **3DMM accuracy.**
> - Since our expression branch solely relies on the 3DMM renderings to provide target expression information, it is indeed affected by the 3DMM's accuracy. When the 3DMM model fails to predict the target expression correctly from the target image, our model replicates the inaccurate expression in the 3D reconstruction as well. We thank the reviewer for pointing out this limitation and we will discuss it in the next revision.
>
> **Social impact.**
> - We have discussed the limitations and the social impact of our work  in details in the supplementary and will move them to the main paper in the next revision.

---

> > ### Comment · Reviewer_FcT2 · 2023-08-21
> >
> > I read the other reviews and the rebuttal. Reviewer PU1D raised some interesting points, especially regarding the warping and the jaw opening artifacts. The authors overall provided satisfactory explanations. I will keep my original rating (WA).

---

### Official Review · Reviewer_ufhJ · 2023-07-13

**Soundness:** 3 good
**Presentation:** 3 good
**Contribution:** 2 fair
**Rating:** 4
**Confidence:** 5

**Summary:**

1. This work proposed a new framework for generalizable head animation in one-shot setting.
2. The results are better than several baselines.

**Strengths:**

1. The target problem is important.
2. The method is reasonable.

**Weaknesses:**

1. Task definition. 1) The practical usage situation need to be clarified? The task is like a 3D version of facial animation. However, the view points changing range is limited and the missing part is not painted. Thus, what is the actual application of this task? Why we should make facial animation 3D-aware when the view point is somewhat limited. 2) The title is miss leading. It looks like a work stressed generalizable ability. However, it is in generalizable ability and 3D-aware. In 2D, there are already a lot work that is generalizable.

2. Comparison. 1) The compared baselines are insufficient. The work in 2D (such as [1,2,3] ) is still needed to be compared, even only in the unchanged view-points as driven videos'. 2) The comparison on CelebV-HQ is necessary. Recently, only HDTF is used in evaluation. However, HDTF is much simpler than CelebV-HQ. The results on a more challenging dataset CelebV-HQ can be much more convincing, both in the comparison on main paper and video.

3. Limitations. Limitations should be discussed in the main paper.

[1] First order motion model for image animation.

[2] Thin-plate spline motion model for image animation.

[3] FNeVR: Neural volume rendering for face animation.

**Questions:**

None.

**Limitations:**

No. Limitations are not discussed in the main paper.

---

> ### Author Rebuttal · Authors · 2023-08-09
>
> **Task definition and 2D baselines.**
> - Please see A1 in the Global Response above and Fig.1 in the rebuttal PDF. We emphasize that compared to the 2D baselines, our method learns a strong 3D geometric prior of human faces and hallucinates more realistic missing parts, especially when the source portrait is captured from an extreme profile view.
> - We will change the title to “Generalizable One-shot 3D Neural Head Avatar” in the next revision.
>
> **Evaluation dataset.**
> - We have discussed the training and evaluation datasets in Section 4.1 of the main paper. Including the baselines added in this rebuttal, we have compared our method to 13 baseline methods in total, each of which is trained with different training datasets. It is infeasible for us to re-train all of them on the same training datasets yet guarantee to achieve their best performance. Thus we follow prior work [1] and carry out evaluation on the *CelebA-HQ* dataset and the testing split of the HDTF dataset. We believe this is a fair comparison since none of the baseline methods, nor our method, has seen these two evaluation datasets during training. Meanwhile, CelebV-HQ is one of our training datasets. Thus, it is not fair to other baselines if we run evaluation on it. We also emphasize that *CelebA-HQ* is a challenging dataset, which includes realistic portrait images with high fidelity details.
>
> [1] Styleheat: One-shot high-resolution editable talking face generation via pre-trained stylegan. ECCV, 2022.
>
> **Limitations.**
> - We have discussed the limitations and the social impact of our work in detail in the supplementary and will move them to the main paper in the next revision.

---

> ### Comment · Area_Chair_h3pV · 2023-08-22
>
> Hi, the reviewer ufhj
>
> Does the rebuttal address your concerns? Any update about your final decision?
>
> Best,
> the AC

---

### Author Rebuttal · Authors · 2023-08-09

### A1. Comparison to 2D baselines (**ufhJ**, **k8VD**)

**Motivation.**

We discuss the benefits of 3D avatars towards talking head synthesis in Lines 35 - 37 and Lines 317 - 319 of the main paper. Other important reasons to study 3D avatars are:
- Human faces are inherently 3D, thus it is physically accurate to model faces in 3D. As shown in Fig.1 in the PDF, compared to 2D baselines, our method has fewer warping artifacts, produces consistent geometry across different views, and is robust to profile view inputs.
- For practical usage, mentioned by **ufhJ**, a 3D avatar is essential in immersive AR / VR applications so that a person can be rendered from novel views to convey eye contact. Meanwhile, 2D methods only change head pose but do not address novel view synthesis easily. Furthermore, traditional pipelines (e.g., gaming) require 3D assets as inputs, which can be extracted from our method.

**Experiments.**

To empirically verify the advantages of our method compared to 2D baselines, we carry out cross-identity reenactment on CelebA-HQ and show results below:

| Method  | CSIM &uarr; | AED &darr;| APD &darr;| FID &darr;| Year|
| ------------- | ------------- | ------------- | ------------- | ------------- | ------------- |
| [1] FOMM  | 0.528  | 0.284 | 0.034 | 37.31 | NeurIPS 2019|
| [2] FaceVid2Vid  | 0.638  | 0.284 | 0.037 | 24.66 | CVPR 2021|
| [3] Thin-Plane | 0.556| 0.272| 0.029| 34.75| CVPR 2022|
| [4] DaGAN | 0.502| 0.276| 0.032| 36.78| CVPR 2022|
| [5] FNeVR | 0.520| 0.283| 0.031| 33.18| NeurIPS 202
| [6] LIA | 0.607| 0.274| 0.037| 34.33| ICLR 2022|
| [7] DPE | 0.635| 0.285| 0.048| 43.38| CVPR 2023|
| [8] MCNET |0.491| 0.262 | 0.030 | 38.89| ICCV 2023|
| Ours | **0.649**| 0.269| **0.018**| **18.68**| |
| Ours* (A3 below) | 0.716| 0.262| 0.019|20.70| |

[1] First order motion model for image animation.

[2] One-Shot Free-View Neural Talking-Head Synthesis for Video Conferencing.

[3] Thin-plate spline motion model for image animation.

[4] Depth-Aware Generative Adversarial Network for Talking Head Video Generation.

[5] FNeVR: Neural volume rendering for face animation.

[6] Latent Image Animator: Learning to Animate Images via Latent Space Navigation.

[7] DPE: Disentanglement of Pose and Expression for General Video Portrait Editing.

[8] Conditioned Memory Compensation Network for Talking Head video Generation.

Overall, our method has better performance. Qualitative results are shown in Fig.1 in the PDF.  In the left example, our method preserves the geometry of the input, while the baselines all "squeeze" the head. In the right example, our method hallucinates realistic missing parts with high fidelity (e.g., wrinkles around the eyes) when the source image is captured from a profile view, while all baselines show warping artifacts.

---
### A2. Jaw opening issue (**8oNf**, **PU1D**)

This issue is caused by the long-tail expression distribution of the training data, rather than being a limitation of our method, (i.e., “mouth wide open” is rare in the training dataset). To verify, we replace EG3D with a deformable 3D GAN [1] for data synthesis, which has two advantages: a) each training pair includes different expressions that encourage the model to produce more accurate deformation. b) the data synthesized by [1] includes larger and more balanced facial deformation. We re-train our method using this dataset and show qualitative results in Fig.4 in the PDF. This model realistically deforms the jaw when the mouth is open, validating our hypothesis. Please also see the video sent to AC.

[1] Next3D: Generative Neural Texture Rasterization for 3D-Aware Head Avatars.

---
### A3. Learning warping in the appearance branch. (**FcT2**, **PU1D**)

**How does the appearance branch ignore source expression?**

- As discussed in the supplementary (Line 105 - 108), we mask out expression-dependent regions (i.e., eyes and mouth) in the source image before it is input into the appearance branch. This way, expression information solely comes from the expression branch.
- As suggested by **8oNf**, we visualize the rendering of $R_v(T_c + T_p)$ in Fig.2 (a) in the PDF, which is a neutral face with intricate details. This shows that $T_p$ adds details while preserving the expression in $T_c$.

**Can we learn warping in the appearance branch?**

- Although the “masking” strategy effectively removes the source expression in the appearance branch, the warping idea suggested by **PU1D** and **8oNf** is inspiring. Interestingly, we found that simply using the appearance encoder to implicitly warp and fuse the appearance and target expressions works best (see Fig.3 in the PDF) as opposed to explicit warping. Specifically, we keep the canonical branch unchanged and merge the appearance and expression branches. This merged branch inputs the concatenation of the source image, canonical rendering, 3DMM rendering with source and target expressions respectively and the source image's mouth mask to an encoder that predicts 2D features. Then it uses the “Lifting” and “Rasterization” operation (Sec. 3.2 in the paper) to produce a tri-plane $T_{ae}$ that encodes both the appearance and the target expression. The intuition is to make the encoder $E_{ae}$ aware of both expressions (from the 3DMM renderings and the mouth mask) and the appearance (from the source image) simultaneously, such that it implicitly learns to transfer details from the source image to proper locations in  $T_{ae}$. We show results in Fig.2 (b) in the PDF and evaluation in the above table (i.e., Ours*). This model has a higher CSIM, because without masking out eyes and mouth in the appearance branch, the model preserves eye and lip color well.  *We emphasize that the “masking” strategy in the paper is sufficient to remove the source expression, but we thank the reviewers for the inspiring warping idea. The preliminary experiment above proves that this idea is feasible and introduces new directions for future work.*

---

### Decision · Program_Chairs · 2023-09-21

**Decision:**

Accept (poster)

**Comment:**

This paper received mixed scores. All reviewers posed some issues and the authors also provided sufficient explanations and additional experimental results. Finally, most of the reviewers are in positive side. Evenly the remaining two reviewers did not change the opinion, they also agree on the acceptance. The AC finally vote to accept, considering 1) the three-branch tri-plane design is novel; 2) the results are SOTA; 3) the results and explanations are sufficient. The authors still need to include all additional results and explanations in the rebuttal to the final revision.